# Re-Evaluation of Chemotherapeutic Potential of Pyoktanin Blue

**DOI:** 10.3390/medicines8070033

**Published:** 2021-06-22

**Authors:** Hiroshi Sakagami, Toshiko Furukawa, Keitaro Satoh, Shigeru Amano, Yosuke Iijima, Takuro Koshikawa, Daisuke Asai, Kunihiko Fukuchi, Hiromu Takemura, Taisei Kanamoto, Satoshi Yokose

**Affiliations:** 1Research Institute of Odontology (M-RIO), Meikai University, 1-1 Keyakidai, Sakado, Saitama 350-0283, Japan; shigerua@dent.meikai.ac.jp; 2Division of Endodontics and Operative Dentistry, School of Dentistry, Meikai University, 1-1 Keyakidai, Sakado, Saitama 350-0283, Japan; to-ko@dent.meikai.ac.jp (T.F.); s-yokose@dent.meikai.ac.jp (S.Y.); 3Division of Pharmacology, Meikai University School of Dentistry, 1-1 Keyakidai, Sakado, Saitama 350-0283, Japan; k-satoh@dent.meikai.ac.jp; 4Department of Oral and Maxillofacial Surgery, Saitama Medical Center, Saitama 350-8550, Japan; yoiijima@saitama-med.ac.jp; 5Department of Microbiology, St. Marianna University School of Medicine, 2-16-1 Sugao, Miyamae-ku, Kawasaki 216-8511, Japan; takuro.koshikawa@marianna-u.ac.jp (T.K.); asai@marianna-u.ac.jp (D.A.); takeh@marianna-u.ac.jp (H.T.); 6Laboratory of Microbiology, Showa Pharmaceutical University, 3-3165 Higashi-Tamagawagakuen, Machida, Tokyo 194-8543, Japan; kanamoto@ac.shoyaku.ac.jp; 7Graduate School of Health Sciences, Showa University, 1-5-8 Hatanodai, Shinagawa-ku, Tokyo 142-8555, Japan; kfukuchi@med.showa-u.ac.jp

**Keywords:** pyoktanin, anticancer activity, oral cancer, apoptosis, chemotherapeutic index, caspase-3, subG1 accumulation, anti-HIV, anti-HSV

## Abstract

**Background:** Pyoktanin blue (PB) is used for staining tissues and cells, and it is applied in photodynamic therapy due to its potent bactericidal activity. However, clinical application of PB as an antiviral and antitumor agent has been limited due to its potent toxicity. For clinical application, the antitumor and antiviral activity as well as the neurotoxicity of PB were re-evaluated with a chemotherapeutic index. **Methods:** Tumor-specificity (TS) was determined by the ratio of CC_50_ against normal oral cells/oral squamous cell carcinoma (OSCC); neurotoxicity by that of normal oral/neuronal cells; antiviral activity by that of mock-infected/virus-infected cells; and potency-selectivity expression (PSE) by dividing TS by CC_50_ (OSCC). **Results:** Antitumor activity of PB (assessed by TS and PSE) was comparable with that of DXR and much higher than that of 5-FU and melphalan. PB induced caspase-3 activation and subG1 cell accumulation in an OSCC cell line (Ca9-22). PB and anticancer drugs showed comparable cytotoxicity against both neuronal cells and OSCC cell lines. PB showed no detectable anti-HIV/HSV activity, in contrast to reverse transferase inhibitors, sulfated glucans, and alkaline extract of leaves of S.P. **Conclusions:** PB showed first-class anticancer activity and neurotoxicity, suggesting the importance of establishing the safe treatment schedule.

## 1. Introduction

Pyoktanin blue (PB), 4-{bis [4-(dimethylamino) phenyl] methylidene}-*N,N*-dimethylcyclohexa-2,5-dien-1-iminium chloride (also known as crystal violet, methyl violet, or gentian violet) is a cationic dye used as a histological stain and visualization [1,2,3,4] and in Gram′s method of classifying bacteria [5] (Figure 1). A paper on the selective bactericidal action of PB dates back to 1912 [6]. PB showed potent fungicidal activity when administered topically, alone or in combination with oral antifungal drugs [7]. Photodynamic therapy with PB effectively eliminated enterococcus faecalis [8] and inhibited biofilm formation [9,10]. PB has been applied to the Valiant^®^ thoracic stent graft system prior to insertion to prevent infection [11]. As one antibacterial mechanism, malfunction of TRX2 by adduct formation with thioredoxin reductase 2 (TRX2) has been suggested [12]. 

On the other hand, publications on the antiviral activity and antitumor activity of PB have been fewer compared with publications on its antibacterial activity. PB showed antiviral activity against Nipah, Hendra virus, their pseudotypes, vesicular stomatitis virus, and human parainfluenza virus type 3, except influenza A virus [13], while another group reported the opposite result, finding that PB inactivated the influenza A virus [14]. As for anticancer activity, PB induced cell death in oral precancerous cells through ROS production, suppressed the progression of DMBA-induced oral precancerous lesions [15], and prompted apoptosis in a human breast cancer cell line [16]. Although many anticancer drugs induce adverse effects, such as oral mucositis, peripheral neurotoxicity, and extravasation at the injection site [17], most of previous studies of PB do not present information of its chemotherapeutic index (safety margin), possible adverse effects, and comparative performance against appropriate antiviral and antitumor agents. In the present study, antitumor activity, neurotoxicity, and antiviral activity of PB were re-evaluated by chemotherapeutic indices and in comparison with appropriate positive controls in order to establish the most effective administration schedule of PB. 

## 2. Materials and Methods

### 2.1. Materials

Pyoktanin blue (PB)(0.2% solution) (MW408) was purchased from Honzo Pharmaceutical Co., Ltd. (Nagoya, Japan); Dulbecco’s modified Eagle’s medium (DMEM) from GIBCO BRL (Grand Island, NY, USA); fetal bovine serum (FBS), doxorubicin (DXR), 3-(4,5-dimethylthiazol-2-yl)-2,5-diphenyltetrazolium bromide (MTT), ribonuclease A, azidothymidine (AZT), 2′,3′-dideoxycytidine (ddC) from Sigma-Aldrich Inc. (St. Louis, MO, USA); dimethyl sulfoxide (DMSO), actinomycin D, dextran sulfate (DS) (5 kDa), 4% paraformaldehyde phosphate buffer solution, propidium iodide from FUJIFILM Wako Chem. (Osaka, Japan); 5-fluorouracil (5-FU) from Kyowa (Tokyo, Japan); curdlan sulfate (79 kDa) (Ajinomoto Co., Inc., Tokyo, Japan); Nonidet-40 (NP-40) from Nakalai Tesque Inc. (Kyoto, Japan); 96-well plates from TPP (Techno Plastic Products AG) (Trasadingen, Switzerland); and 100 mm dishes from True Line (Nippon Genetics Co., Ltd., Tokyo, Japan).

### 2.2. Cell Culture

Human oral squamous cell carcinoma (OSCC) cell lines (Ca9-22 (derived from gingiva), HSC-2, HSC-3, HSC-4 (derived from tongue)), rat adrenal pheochromocytoma cell line (PC12), human neuroblastoma cell line (SH-SY5Y), and rat Schwann cell line (LY-PPB6) (purchased from RIKEN Cell Bank, Tsukuba, Japan) were cultured at 37 °C in DMEM supplemented with 10% heat (56 °C, 30 min)-inactivated FBS, 100 U/mL penicillin G, and 100 µg/mL streptomycin sulfate under a humidified 5% CO_2_ atmosphere. 

We previously established human gingival fibroblast (HGF), periodontal ligament fibroblast (HPLF), and pulp cell (HPC) from the first premolar extracted tooth and periodontal tissues of a twelve-year-old girl, according to the guideline of Institutional Board of Meikai University Ethic Committee (No. A0808) [18]. Since these cells have a limited lifespan [18], cells at 10–18 population doubling level (PDL) were used in the present study.

Differentiated PC12 cells were prepared by the “overlay method”, as described previously [19]. In brief, PC12 cells were cultured in the serum-free DMEM supplemented with 50 ng/mL NGF, and overlayed with fresh NGF solution at Day 3. The Day 6 cells with extended neurites were used for the experiment. 

### 2.3. Assay for Cytotoxic Activity

Cells were detached by trypsin, inoculated at 2 × 10^3^ cells/0.1 mL in a 96-microwell plate and incubated for 48 h to recover from the trypsin damage and to secure complete cell attachment. Microscopical observation revealed that at this time, the OSCC cells occupied approximately 20–30% of the bottom of each well. Then, the culture medium was replaced with 0.1 mL of fresh medium containing different concentrations of test compounds (PB (0.2, 0.4, 0.8, 1.5, 3, 6, 12, and 25 μM); DXR (0.08, 0.16, 0.31, 0.63, 1.25, 2.5, 5, and 10 μM); 5-FU (8, 16, 31, 63, 125, 250, 500, and 1000 μM); melphalan (2, 4, 8, 16, 31, 63, 125, 250 μM); and vehicle (DMSO) (0.008, 0.016, 0.031, 0.063, 0.125, 0.25, 0.5, and 1%)). Cells were then incubated further for 48 h and the relative viable cell number was then determined by the MTT method, as described previously [20]. In brief, cells were incubated for 2 h with 0.2 mg/mL MTT, and formazan precipitate was dissolved with 100 μL DMSO; the absorbance of the cell lysate at 560 nm (that reflects the relative viable cell number) was then determined using a microplate reader (Infinite F50R; TECAN, Männedorf, Switzerland). Cytotoxicity caused by the vehicle (DMSO) was subtracted. The concentration of compound that reduced the viable cell number by 50% (CC_50_) was determined from the dose–response curve, and the mean value of CC_50_ for each cell type was calculated from triplicate assays. To test the reproducibility, cytotoxicity assays were repeated three times.

The absorbance value at 560 nm of control OSCC cells 96 h after inoculation reached approximately 1.0, within the range of linearity of the absorbance. We used the 48 h treatment time to make the cells go around the cell cycle twice for the calculation of CC_50_ value. When cytostatic agents such as 5-FU are used, this consideration is necessary. 

### 2.4. Calculation of Tumor-Specificity Index (TS)

TS was calculated using the following equation: TS = mean CC_50_ against three normal human oral mesenchymal cells (HGF + HPLF + HPC)/mean CC_50_ against four OSCC cell lines (Ca9-22 + HSC-2 + HSC-3 + HSC-4), as shown by D/B in Table 1. Since both Ca9-22 and HGF cells were derived from gingival tissue [21], the relative sensitivity of these cells was also compared (as shown by C/A in Table 1).

### 2.5. Calculation of Potency-Selectivity Expression (PSE)

Parameters that reflect both tumor-specificity (TS) and cytotoxicity against tumor cells (reciprocal of CC_50_ value) are useful for the treatment of cancer patients. PSE was calculated using the following equation: PSE = 100 × TS/CC_50_ (tumor cells) (100 × D/B^2^) (three normal oral cells vs. four OSCC cell lines) and 100 × C/A^2^ (HGF vs. Ca9-22) (Table 1).

### 2.6. Cell Cycle Analysis

Treated and untreated Ca9-22 cells (approximately 10^6^ cells) were harvested, fixed with 1% paraformaldehyde, treated with 0.2 mg/mL RNase A (preheated for 10 min at 100 °C to inactivate DNase), stained for 15 min with 0.01% propidium iodide in the presence of 0.01% NP-40 to prevent cell aggregation, filtered through Falcon^®^ cell strainers (40 μM) (Corning, NY, USA), subjected to cell sorting (SH800 Series; SONY Imaging Products and Solutions Inc., Kanagawa, Japan), and then analyzed with Cell Sorter Software version 2.1.2. (SONY Imaging Products and Solutions Inc.), as described previously [20].

### 2.7. Western Blot Analysis

Control and treated Ca9-22 cells at or near confluent phase were collected and lysed, and protein samples of cell lysates (15 μg) were applied to SDS-polyacrylamide gel electrophoresis. After electrophoresis, the separated proteins were transferred onto a PVDF filter. The blots were blocked in skim milk and then probed for 120 min with a primary antibody cocktail (1:250) from Apoptosis Western Blot Cocktail kit (purchased from Abcam, Cambridge, UK). The blots were washed, then probed with horseradish peroxidase conjugated secondary antibody cocktail (1:100). Immunoreactivities were determined using Amersham ECL Select. Images were acquired using the ChemiDoc MP System and Image Lab 4.1 software (Bio-Rad Laboratories), as described previously [22].

### 2.8. Assay for Anti-Human Immunodeficiency Virus (HIV) Activity

Human T-cell leukemia virus I (HTLV-I)-bearing CD4-positive human T-cell line MT-4 [23] was cultured in RPMI-1640 medium supplemented with 10% FBS and infected with HIV-1_IIIB_ at a multiplicity of infection (MOI) of 0.01. HIV- and mock-infected MT-4 cells (3 × 10^4^ cells/96-microwell) were incubated for five days with different concentrations of samples in triplicate, and the relative viable cell number was determined by MTT assay, as described previously [24]. Under this condition, cell viability of HIV-infected cells reached to nearly 0% (baseline). Thus, the concentration that reduced the viable cell number of the mock-infected cells to 50% of untreated control cells (CC_50_) and the concentration that recovered the viable cell number of the HIV-infected cells to 50% of the control cells from the baseline (EC_50_) were determined from the dose–response curve. The anti-HIV activity was evaluated by the selectivity index (SI), calculated using the following equation: SI = CC_50_/EC_50_.

### 2.9. Assay for Anti-Herpes Simplex Virus (HSV) Activity

HSV-1 strain F was cultured in MEM supplemented with 10% FBS. Anti-HSV activity was measured by method 3, as described previously [24]. In brief, 100 × HSV (MOI = 1) was exposed to serially diluted PB for 3 min, diluted × 100 to bring MOI = 0.01, added to African green monkey kidney Vero cells, and then incubated for 3 days. Mock-infected Vero cells were treated for 3 min with the serially diluted PB for 3 min, washed, and then incubated for 3 days. The relative viable cell number was determined by the MTT method. From the dose–response curve, 50% cytotoxic concentration (CC_50_) in mock-infected cells and the 50% effective concentration (EC_50_) in HSV-infected cells were determined. Since HSV-infection did not reduce the viability to zero %, two values of EC_50_ were determined. EC_50_-I was determined as the concentration at which the viability was restored to the midpoint between that of HSV-infected cells and that of mock-infected cells. EC_50_-II was determined as the concentration at which the viability was restored to 50% of that of the mock-infected cells. The anti-HSV activity was evaluated by the following selectivity indices (SI-I and SI-II): SI-I = CC_50_/EC_50_-I; SI-II = CC_50_/EC_50_-II [24].

### 2.10. Statistical Treatment

Each value represents the mean ± S.D. of triplicate assays. Significance between control and sample was evaluated by one-way analysis of variance (ANOVA) for multiple comparisons.

## 3. Results

### 3.1. Tumor-Specificity of Pyokanin (PB)

Pyoktanin (PB), doxorubicin (DXR), and melphalan were cytotoxic, reducing the cell viability to baseline (0%), while 5-FU was cytostatic against four human oral squamous cell carcinoma (OSCC) cell lines (Ca9-22, HSC-2, HSC-3, HSC-4), not completely killing out the cells. All of them showed higher cytotoxicity against four OSCC cell lines than three normal oral cells (HGF, HPLF, HPC). These data were reproducible in three independent experiments (Figure 2). From the dose–response curve, 50% cytotoxic concentration (CC_50_), tumor-specificity (TS = mean CC_50_ (normal cells)/mean CC_50_ (OSCC)), and potency-selectivity expression ((PSE = 100 x TS/mean CC_50_ (OSCC)) were calculated (Table 1). PB showed TS and PSE values ((30.8 + 26.6 + 25.4)/3 = 27.6 and (6722 + 9203 + 8067)/3 = 7997) slightly lower than those of doxorubicin ((30.0 + 40.7 + 33.5)/3 = 34.7 and (9986 + 16,525 + 11,192)/3 = 12,568) but much higher than 5-FU ((3.1 + 3.0 + 1.8)/3 = 2.6 and (0.9 + 0.9 + 0.3)/3 = 0.7) and melphalan ((11.8 + 10.3 + 9.4)/3 = 10.5 and (70 + 57 + 46)/3 = 58).

Next, tumor-specificity was investigated with Ca9-22 and HGF cells, both prepared from gingival tissue. This time, PB showed TS and PSE values ((40.0 + 43.5 + 37.9)/3 = 40.5 and (13,197 + 22,734 + 19,850)/3 = 18,594) slightly higher than those of doxorubicin ((24.4 + 22.1 + 29.9)/3 = 25.5 and (5930 + 4902 + 8964)/3 = 6599) but much higher than those of 5-FU ((8.1 + 3.2 + 7.4)/3 = 6.2 and (6.5 + 1.0 + 5.4)/3 = 4.3) and melphalan ((7.4 + 5.2 + 5.2)/3 = 5.9 and (27 + 13 + 14)/3 = 18) (Table 1).

PB induced cell shrinkage (A), caspase-3 activation (assessed by cleavage of procaspase 3 and PARP) (B), and subG1 accumulation (C) to a similar or slightly lower extent than that attained by actinomycin D (Figure 3). These results suggest the induction of apoptosis by pyoktanin. However, such an effect of PB only became significant (*p* < 0.05, vs. control, ANOVA) at 2.2 μM but not at 0.74 μM, indicating the optimal range of apoptosis is very narrow.

### 3.2. Potent Neurotoxicity of PB

Micromolar concentration of PB and DXR induced potent cytotoxicity against both undifferentiated (rat adrenal pheochromocytoma PC12, human neuroblastoma SH-SY5Y) (Figure 4A) and differentiated neuronal cells (PC12) with extended neurites [19] (Figure 4B). On the other hand, neurotoxicity of 5-FU and melphalan were observed at higher concentrations. From the dose–response curve of Figure 4, the CC_50_ value of PB and anticancer drugs were determined (Table 2). It is clear that PB was more cytotoxic to all neuronal cells, either undifferentiated (CC_50_ = 0.179, 0.162, 0.132 μM) or differentiated (CC_50_ = 0.252 μM) than OSCC (CC_50_ = 0.36 μM). By dividing the CC_50_ (OSCC) by the mean CC_50_ (neuronal cells), PB was found to show approximately 2.0-fold higher cytotoxicity against neuronal cells than OSCC. Similarly, DXR, 5-FU and melphalan showed approximately 1.8-, 3.6-, and 1.4-fold higher cytotoxicity against neuronal cells than OSCC (Table 2).

### 3.3. Re-Evaluation of Antiviral Activity of PB

#### 3.3.1. Pyoktanin (PB) failed to induce anti-HIV activity

Pyoktanin was highly cytotoxic to the MT-4 cell line, a target cell used for measuring anti-HIV activity. Its cytotoxicity was detectable above 0.31 μM. HIV infection (MOI) reduced the viability of MT-4 to zero. PB recovered the cell viability only to 15.7% (A), whereas inhibitors of HIV′s reverse transcriptase such as AZT (B) and ddC (C) as well as sulfated polysaccharides such as dextran sulfate (D) and curdlan sulfate (E) recovered the cell viability up to 85.1, 68.7, 91.0, and 111.3% of control, respectively (Figure 5).

From the dose–response curve of mock-infected and HIV-infected cells, 50% cytotoxic concentration (CC_50_) and 50% protective concentration (EC_50_) were determined, respectively. Anti-HIV activity was quantitated by the selective index (SI), calculated by dividing CC_50_ by EC_50_ (Table 3). It was found that PB did not show detectable anti-HIV activity (SI < 1), in contrast to AZT (SI = 5082), ddC (SI = 1340), dextran sulfate (SI = 4200), and curdlan sulfate (SI > 5294).

#### 3.3.2. Pyoktanin (PB) failed to induce anti-HSV activity

We recently demonstrated that alkaline extract of leaves of *Sasa* sp. (SE) rapidly inactivated HSV as well as HIV [24]. Therefore, the anti-HSV activity of PB was next investigated in a short exposure system, using alkaline extract of leaves of *Sasa* sp. (SE) as positive control [25]. HSV (MOI = 1) was first exposed for 3 min to the indicated concentrations of PB (A) or SE (B), followed by 100x dilution to bring the MOI to 0.01, and was then added to Vero cells. After incubation for 3 days, the cell viability was reduced to 26% (A) and 15.6% (B) of control (Figure 6). When PB (0.25–245 μM) and SE (0.1–60 mg/mL) were added during a short exposure time, only SE (A) but not PB (B) recovered the cell viability above control (Figure 6).

From the dose–response curve of mock-infected and HSV-infected cells, 50% cytotoxic concentration (CC_50_) and 50% protect concentration (EC_50_) were determined, respectively (Figure 6B). Since HSV infection could not reduce the viability to zero (Figure 6B), in contrast to HIV infection (Figure 5), EC_50_-I and EC_50_-II values were determined. Anti-HSV activity was quantitated by the selective index (SI), calculated by dividing CC_50_ by EC_50_-I or EC_50_-II (Table 3). It was found that PB did not show detectable anti-HIV activity (SI-I < 0.02, SI-II < 0.02; maximum cell recovery = 29.7%), in contrast to SE (SI-I > 76.9, SI-II > 89.6; maximum cell recovery = 101.4%) (Table 4).

## 4. Discussion

We used human OSCC cell lines and human mesenchymal normal oral cells rather using human epithelial cells for determining anticancer activity (based on TS and PSE) since most of anticancer drugs show potent cytotoxicity against epithelial cells. Using the present system, we previously confirmed that many anticancer drugs show excellent TS and PSE values [26,27].

The present study demonstrated that pyoktanin (PB) showed potent anti-cancer activity against four human oral squamous cell carcinoma (OSCC) cell lines. Its tumor-specificity (TS = 27.6(D/B), 40.5(C/A); PSE = 7997(100D/B^2^), 18,594(100C/A^2^)) was comparable with doxorubicin (TS = 34.7, 25.5; PSE = 12,568, 6599) and much higher than 5-FU (TS = 2.6, 6.2; PSE = 0.7, 4.3) and melphalan (TS = 10.5, 5.9; PSE = 58, 18).

Furthermore, PB induced apoptotic cell death (characterized by cell shrinkage, caspase-3 activation, and subG1 cell accumulation) in the Ca9-22 cell line prepared from gingiva, further adding the antitumor potential of PB. A previous study demonstrated that PB induced significant cell death in oral precancerous cells via decreased phosphorylation of p53(Ser15) and NFκB (Ser536) in vitro and oral-base-formulated PB effectively suppressed the progression of DMBA-induced oral precancerous lesions in vivo [15]. Taken together, PB may be applicable as a therapeutic drug not only for oral malignant disorder but also for OSCC.

Previous studies showed a lower bound on the virtually safe dose (VSD) of PB to be 2 ppb (4.9 nM) for female mice and 1 ppb (2.45 nM) for male mice [28]. Administration of PB (600 ppm = 1.47 mM, 24 months) induced hepatocellular adenoma at the probability of 4/89, whereas the induction of thyroid gland follicular cell adenoma or adenocarcinoma was rare at 300~600 ppm (0.735~1.47 mM for 24 months) [29]. PB, having log P of 4.488, can be easily permeated through the plasma membrane. PB binds to DNA, and together with its cellular toxicity, complicates both the testing of PB in vitro and the interpretation of the results. PB around 5 μM seems to be the borderline of carcinogenicity [29]. This concentration is 1 order higher than that of CC_50_ of PB for four OSCC cell lines (Figure 2).

The present study also demonstrated that PB induced potent neurotoxicity at nearly half the concentration required to inhibit OSCC growth (Figure 4). We found that PB inhibited the neurite outgrowth, in similar fashion to bortezomib (1 ng/mL) [20], X-ray (506 mGy) [30], paclitaxel (5 nM) [31], and Aβ_1-42_ (20 nM) [32]. We also found that during the differentiation of PC12 toward maturing neuronal cells, the sensitivity against PB, DXR, 5-FU, and melphalan was reduced 1.4- (=0.252/0.179), 3.1- (=0.215/0.069), 21.2- (=297/14), and 3.4-fold (26.4/7.72), respectively (Table 2). This is consistent with our previous finding that the sensitivity of PC12 cells toward docetaxel, SN-38 (active component of irinotecan), etoposide, gefinitib, DXR, melphalan, 5-FU, and methotrexate was reduced to various extents (maximum: 10,000-fold) during NGF-induced differentiation [33]. Chemotherapy-induced peripheral neuropathy (CIPN) is not recovered from quickly by a drug withdrawal, and some disorders may remain for a lifetime. The incidence of CIPN 1, 3, and 6 months after chemotherapy is reported to be 68.1, 60.0, and 30.0%, respectively [34]. Platinum, taxane, and vinca alkaloid are known as causative agents of CIPN. Cisplatin is cumulative and causes sensorineural deafness in the high range due to acoustic nerve damage [35]. The incidence of oral mucositis has been reported to be 5–50% among patients receiving standard-dose chemotherapy and 68–98% among patients receiving high-dose chemotherapy for hematopoietic stem cell transplantation [36]. Painful oral mucositis lowers the patient’s QOL, and oral intake of nutrition and may trigger systemic infection by breaking the gateway function of oral mucosa. At present, there are no established preventive or curative methods for oral mucositis [37]. When cytotoxic chemotherapy drugs are accidentally leaked at the injection site, they can cause multiple emergencies by local and systemic reactions [38,39]. Vesicants such as doxorubicin (classified as anthracycline) and melphalan (classified as alkylating agent) bind to DNA and induce the formation of blisters and/or cause tissue destruction, while irritants such as 5-FU (classified as antimetabolite) can cause pain at the injection site or along the vein [17]. Considering the high incidence of side effects of anticancer drugs, it is necessary to investigate the extent to which PB induces oral stomatitis and extravasation in comparison with anticancer drugs. Our preliminary experiment demonstrated that the cytotoxic action of PB was so quick that shortening the exposure time up to 2 min dramatically reduced the cytotoxicity of PB against HGF and HPLF (Appendix A). For clinical application, such countermeasures may be effective.

Oral squamous cell carcinoma (OSCC) is the most common malignant tumor of the oral cavity. Human papilloma virus (HPV) has been proposed as a risk factor in OSCC development. Epstein-Barr virus and HSV Type 1 have been proposed to be involved in oral carcinogenesis, albeit in the absence of convincing evidence [40]. HIV could play a role in HPV-associated pathogenesis by exerting oncogenic stimulus via transactivator protein (Tat) [41]. Therefore, we thought if PB were to inhibit HSV and HIV, PB treatment might be efficacious in reducing the incidence of OSCC. In contrast to previous papers that demonstrated the antiviral potential of PB [13,14], we failed to show anti-HIV and anti-HSV activity in this study. The reason for the discrepancy between our findings and theirs may be differences in the type of viruses, culture conditions, or calculation method of the chemotherapy index.

In conclusion, the present study demonstrated that PB induces potent antitumor activity against OSCC (as evidenced by TS and PSE value, almost equivalent with those of anticancer drugs), in addition to its antibacterial, antifungal, antihelminthic, and antitrypanosomal activities. However, PB also showed higher neurotoxicity against three neuronal cells, either undifferentiated or differentiated. We previously reported the tumor-specificity (TS = 10.4(D/B) + 4.6(C/A) = 15; PSE = 1551(100D/B^2^) + 246 (100C/A^2^) = 1797) [27] and neurotoxicity of cisplatin (B/E = 0.73/4.4 = 0.17) [20]. Compared with these values, the tumor-specificity of PB (TS = 30.8 + 40.0 = 70.8; PSE = 6722 + 13197 = 19917) (Table 1) was 4.7-fold (70.8/15)~11.1-fold (19919/1797) that of cisplatin. On the other hand, the neurotoxicity of PB (B/E = 2.0) was 11.8-fold that of cisplatin (B/E = 4.4/0.73 = 0.17). This further supports the notion that PB is first-class as both an anticancer drug and a neurotoxic agent, with these two activities tightly coupled.

Considering its carcinogenicity by interaction with DNA, the combination of PB with appropriate photodynamic therapy may be the safest clinical application in the oral cavity.

## Figures and Tables

**Figure 1 medicines-08-00033-f001:**
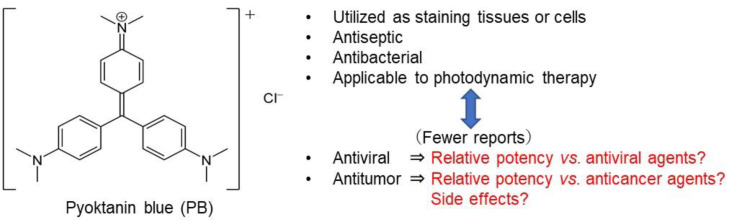
Structure of pyoktanin blue (PB) and design of the present study.

**Figure 2 medicines-08-00033-f002:**
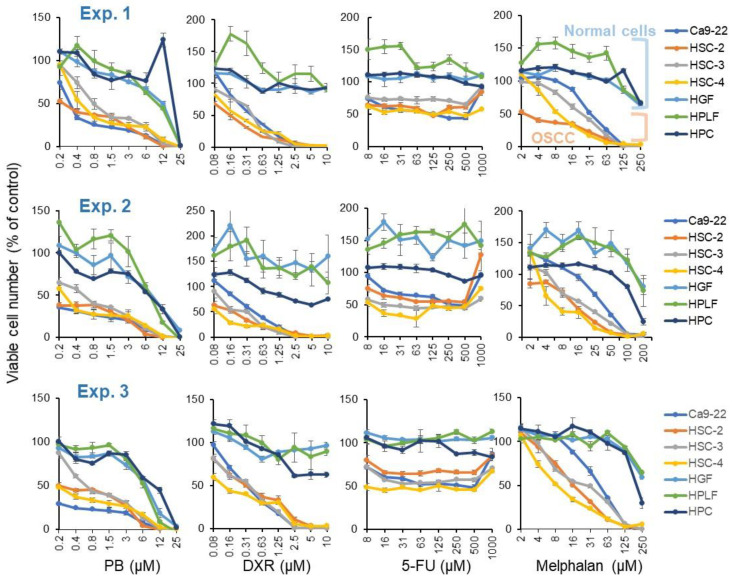
Dose–response curve of cytotoxicity of pyoktanin (PB) and three anticancer drugs (DXR, 5-FU and melphalan). Cells were incubated for 48 h with the indicated concentrations of test compounds. Experiments were repeated three times. Each value in each panel represents mean ± S.D. of triplicate assays. PB, pyoktanin blue; DXR, doxorubicin; 5-FU, fluorouracil.

**Figure 3 medicines-08-00033-f003:**
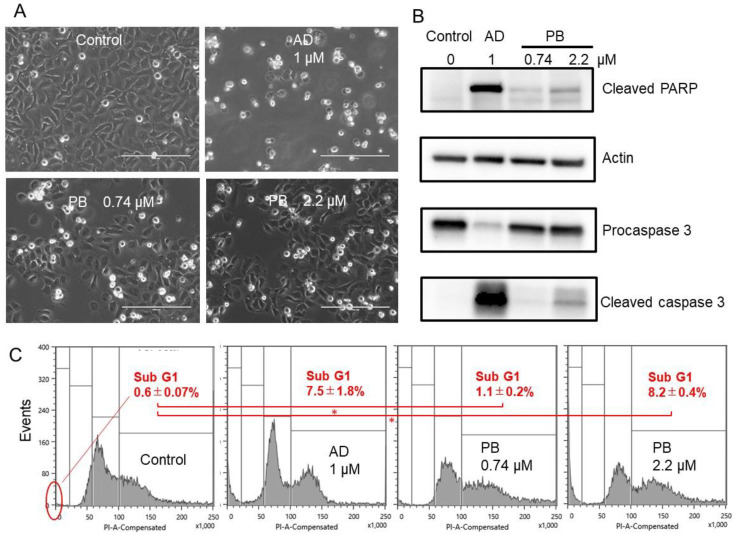
Pyoktanin (PB) induces apoptosis in Ca9-22 cells. Ca9-22 cells were incubated for 24 h and then subjected to morphological observation under the light microscopy (**A**), Western blot analysis (**B**), and cell sorter analysis (**C**). The percentage of subG1 population was determined in triplicate and expressed as mean ± S.D. AD, actinomycin D; * statistically significant difference from control (*p* < 0.05).

**Figure 4 medicines-08-00033-f004:**
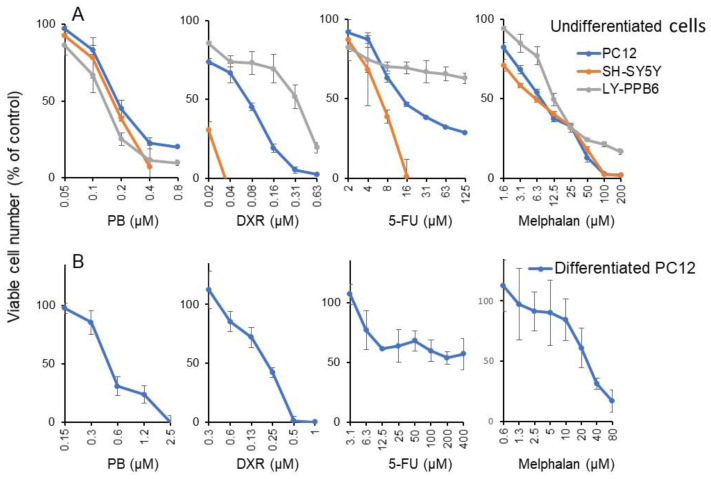
Cytotoxicity of PB and anticancer drugs against PC12, SH-SY5Y, and LY-PPB6 (**A**) and differentiated PC12 cells (**B**). Each value represents mean ± S.D. of triplicate assays.

**Figure 5 medicines-08-00033-f005:**
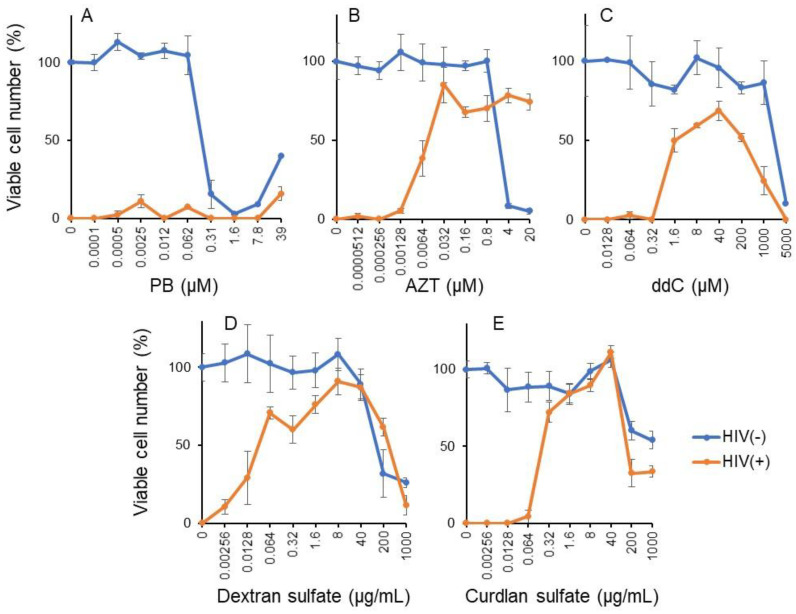
PB failed to induce anti-HIV activity. Mock and HIV-infected MT-4 were treated with the indicated concentrations of pyoktanin (**A**), AZT (**B**), ddC (**C**), dextran sulfate (**D**), and curdlan sulfate (**E**) for 5 days to determine the viability by MTT method. Each value represents mean ± S.D. of triplicate assay.

**Figure 6 medicines-08-00033-f006:**
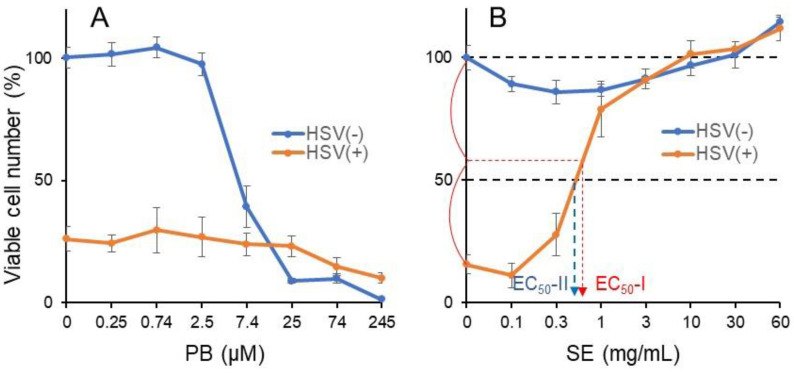
PB failed to induce anti-HSV activity. 100xHSV (MOI = 1) and the indicated concentrations of PB (**A**) or SE (**B**) were mixed for 3 min, diluted to x100 with culture medium to bring MOI = 0.01, added to Vero cells, and then incubated for 3 days (HSV(+)). Mock-infected Vero cells were treated for 3 min with the indicated concentrations of sample without virus, washed, and cultured for 3 days in fresh culture medium (HSV(−)) according to method 3 [24]. Viable cell number was then determined by MTT method. Each value represents mean ± S.D. of triplicate assay.

**Table 1 medicines-08-00033-t001:** TS and PSE values of PB and anticancer drugs, determined by three independent experiments.

	CC_50_ (μM)												
	Human Oral Squamous Carcinoma Cell					Human Normal Oral Cells				TS		PSE	
	Ca9-22	HSC-2	HSC-3	HSC-4	Mean	HGF	HPLF	HPC	Mean				
	A				B	C			D	D/B	C/A	100D/B^2^	100C/A^2^
PB	0.30	0.26	0.81	0.46	0.46	12.1	10.5	19.6	14.1	30.8	40.0	6722	13,197
DXR	0.41	0.16	0.43	0.20	0.30	10.0	7.1	10.0	9.0	30.0	24.4	9986	5930
5-FU	124	109	1000	73	326	1000	100	1000	1000	3.1	8.1	1	7
Mel	27	12	19	9	17	200	200	200	200	11.8	7.4	70	27
PB	0.19	0.19	0.54	0.23	0.29	8.3	7.6	7.2	7.7	26.6	43.5	9203	22,734
DXR	0.45	0.18	0.27	0.09	0.25	10.0	10.0	10.0	10.0	40.7	22.1	16,525	4902
5-FU	314	1000	20	9	336	1000	1000	1000	1000	3.0	3.2	1	1
Mel	39	11	18	5	18	200	200	155	185	10.3	5.2	57	13
PB	0.19	0.24	0.61	0.23	0.32	7.3	6.4	10.4	8.0	25.4	37.9	8067	19,850
DXR	0.33	0.34	0.38	0.14	0.30	10.0	10.0	10.0	10.0	33.5	29.9	11,192	8964
5-FU	136	1000	1000	31	542	1000	1000	1000	1000	1.8	7.4	0.3	5
Mel	38	13	23	6	20	200	200	165	188	9.4	5.2	46	14

The 50% cytotoxic concentration (CC_50_) was determined by Figure 2. TS (D/B) was determined by dividing [(CC_50_ (HGF) + CC_50_ (HPLF) + CC_50_ (HPC))/3] by [(CC_50_ (Ca9-22) + CC_50_ (HSC-2) + CC_50_ (HSC-3) + CC_50_ (HSC-4))/4]. TS (C/A) was determined by dividing CC_50_ (HGF) by CC_50_ (Ca9-22). PSE was determined by dividing TS (D/B or C/A) by B or A and then multiplying by 100. PB, pyoktanin blue; Mel. melphalan.

**Table 2 medicines-08-00033-t002:** PB and anticancer drugs were more toxic to neuronal cells than OSCC cell lines.

	CC_50_ (μM).						
	Neuronal Cells						
	PC12	SH-SY5Y	LY-PPB6	Diff.PC-12	Mean	OSCC	Neurotoxicity
					E	B	B/E
PB	0.179	0.162	0.132	0.252	0.181 ^1^	0.36 ^2^	2.0
DXR	0.069	0.02	0.323	0.215	0.157	0.28	1.8
5-FU	14	5.85	125	297	110	401	3.6
Melphalan	7.72	5.89	12.3	26.4	13.1	18.3	1.4

^1^ determined from Figure 4. ^2^ mean of three experiments, calculated from Table 1.

**Table 3 medicines-08-00033-t003:** Quantification of anti-HIV activity of pyoktanin.

	CC_50_ (µg/mL)	EC_50_ (µg/mL)	SI
PB	0.168 μM	>39 μM	<1
Positive controls:			
AZT	48.3 μM	0.0095 μM	5082
ddC	2163 μM	1.61 μM	1340
Dextran sulfate	120 μg/mL	0.0286 μg/mL	4200
Curdlan sulfate	>1000 μg/mL	0.189 μg/mL	>5294

**Table 4 medicines-08-00033-t004:** Quantification of anti-HSV activity of pyoktanin.

Test Sample	Viability of HSV-Infected Cells	CC_50_	EC_50_-I	EC_50_-II	Anti-HSV Activity		Max. Cell Recovery
					SI-I	SI-II	
PB	26.0%	5.88 μM	>245 μM	>245 μM	<0.02	<0.02	29.7%
Positive control:							
SE	14.2%	>60 mg/mL	>0.78 mg/mL	0.67 mg/mL	>76.9	>89.6	101.4%

Selective index (SI-I, SI-II) was determined by dividing CC_50_ by EC_50_-I or EC_50_-II (Figure 6)

## Data Availability

We provide our unpublished data in support of our findings in Appendix A.

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
