# Peer review of "Re-Evaluation of Chemotherapeutic Potential of Pyoktanin Blue"

_medicines, 2021, doi:10.3390/medicines8070033_

Round 1
Reviewer 1 Report
This manuscript investigates the use of Pyoktanin Blue (PB ) in affecting the viability of selected Authors' cancer cell lines. In general, I believe that this paper would be of interest to the readers of Medicines. However, additional results and discussion is necessary to describe what is shown by the data. As it currently stands, the manuscript has not enough data to present results of the cell viability reduction mechanism by nanocomposite. Please additional, more specific comments below.
Part Materials and Methods must be improved and reorganized. In a recent stage, there are several unknowns including methods of treatment, cell preparation, number of repetitions.
In part 2.3. Assay for cytotoxic activity there is no information about used anticancer drugs including DXR, 5-FU, Melphalan and used concentration, time of exposition, and treatment details.
Line 98-100: „After 48 h, the medium was replaced with 0.1 mL of fresh medium containing different concentrations of test compounds. Cells were incubated further for 48 h and the relative viable cell “ 96 hours of cell growth on 96 well plate is a rather long time for cultivation, especially for cancer cell lines. Please justify the selection of the time parameter
Line 120: “Treated and untreated Ca9-22 cells “there is no information about the treatments reagent and concentrations
Line 164-166: “Statistical treatment. Each value represents the mean ±S.D. of triplicate assays. Significance between control and sample was evaluated by paired student t-test.” Two-sample t-test (is a method used to test whether the unknown population means of two groups are equal or not. I recommended recalculating the data and use a multiple comparison method like analysis of variance (ANOVA).
Figure 3. there is a lack of abbreviation explanation of AD, Act.D.
Figure 4. there is a lack of results of melphalan treatment of undifferentiated cells.
It would have been interesting to see if/how cell morphology changed in undifferentiated and differentiated cell lines after treatment by PB, melphalan, DXR, and 5-FU comparing to the control – non-treated group.
The authors did not present the goal of this study and the obtained results did not fully support the potential anticancer activity of PB.
Editing grammar could help improve readability. I would recommend review by a native English speaker, if possible
Author Response
Response to Reviewer 1
Reviewer 1:
This manuscript investigates the use of Pyoktanin Blue (PB ) in affecting the viability of selected Authors' cancer cell lines. In general, I believe that this paper would be of interest to the readers of Medicines. However, additional results and discussion is necessary to describe what is shown by the data. As it currently stands, the manuscript has not enough data to present results of the cell viability reduction mechanism by nanocomposite. Please additional, more specific comments below.
Response: Thank you for constructive comments. I respond your comments as described below.
Part Materials and Methods must be improved and reorganized. In a recent stage, there are several unknowns including methods of treatment, cell preparation, number of repetitions.
In part 2.3. Assay for cytotoxic activity there is no information about used anticancer drugs including DXR, 5-FU, Melphalan and used concentration, time of exposition, and treatment details.
Response: We have added the information of dose of anticancer drugs in line 103~105, and treatment details in line 98-118.
Line 98-100: „After 48 h, the medium was replaced with 0.1 mL of fresh medium containing different concentrations of test compounds. Cells were incubated further for 48 h and the relative viable cell “ 96 hours of cell growth on 96 well plate is a rather long time for cultivation, especially for cancer cell lines. Please justify the selection of the time parameter
Response:
We have justified the selection of the time parameter by stating in the text that:
- Cells were detached by trypsin, inoculated at 2×103 cells/0.1 mL in a 96-microwell plate, and incubated for 48 h to recover from the trypsin damage and secure the complete cell attachment. Microscopical observation revealed that at this time, the OSCC cells occupied approximately 20-30% of the bottom of each well (line 98-101).
- The absorbance value at 560 nm of control OSCC cells 96 h after inoculation reached approximately 1.0, within the range of linearity of the absorbance. We have used the 48 h treatment time to make the cells go around the cell cycle twice for the calculation of CC50 When cytostatic agents such as 5-FU are used, this consideration is necessary.” (line 114-118).
Line 120: “Treated and untreated Ca9-22 cells “there is no information about the treatments reagent and concentrations
Response:
We added the information about the treatments reagent (line 74-77) and concentrations (line 132-135).
Line 164-166: “Statistical treatment. Each value represents the mean ±S.D. of triplicate assays. Significance between control and sample was evaluated by paired student t-test.” Two-sample t-test (is a method used to test whether the unknown population means of two groups are equal or not. I recommended recalculating the data and use a multiple comparison method like analysis of variance (ANOVA).
Response:
You are right. We corrected the paired student t-test to one-way analysis of variance (ANOVA) (line 179-180) and recalculated. When we applied this method, we confirmed that PB-induced apoptosis only at 2.2μM, but not at 0.74 μM, indicating the optimal range of apoptosis is very narrow (line 216-218).
Figure 3. there is a lack of abbreviation explanation of AD, Act.D.
Response:
Thank you for pointing out. We have unified AD and Act. D. into AD, and then added the explanation of AD in the legend of Figure 3.
Figure 4. there is a lack of results of melphalan treatment of undifferentiated cells.
Response:
We added the cytotoxicity of melphalan treatment of undifferentiated cells in Figure 4. From the dose-response curve, the CC50 values for melphalan for undifferentiated PC12, SH-SY5Y and LY-PPB6 were determined, and these values were added to Table 2.
It would have been interesting to see if/how cell morphology changed in undifferentiated and differentiated cell lines after treatment by PB, melphalan, DXR, and 5-FU comparing to the control – non-treated group.
Response:
We found that PB inhibited the neurite outgrowth, in similar fashion with bortezomib (1 ng/mL) [20], X-ray (506 mGy) [30]. paclitaxel (5 nM) [31] and Aβ1-42 (20 nM)[32]. We also found that during the differentiation of PC12 toward maturing neuronal cells, the sensitiv-ity against PB, DXR, 5-FU and melphalan was reduced 1.4 (=0.252/0.179), 3.1 (=0.215/0.069), 21.2-fold (=297/14) and 3.4-fold (26.4/7.72), respectively (Table 2). This is consistent with our previous finding that that sensitivity of PC12 cells toward docetaxel, SN-38 (active component of irinotecan), etoposide, gefinitib, DXR, melphalan, 5-FU, meth-otrexate) reduced to various extents (maximum: 10,000-fold) during NGF-induced differ-entiation [33].
These statements were added (line 347-355).
The authors did not present the goal of this study and the obtained results did not fully support the potential anticancer activity of PB.
Response:
We first described the rational of the present anti-tumor assay based on TS and PSE using human OSCC cell lines and human mesenchymal normal oral cells rather using human epithelial cells, since most of anticancer drugs shows potent cytotoxicity against epithelial cells. keratinocyte toxicity. Using the present system, we confirmed that many anticancer drugs show excellent TS and PSE values [26, 27] (line 318-322)
Editing grammar could help improve readability. I would recommend review by a native English speaker, if possible
Response
We checked English thoroughly.

Reviewer 2 Report
A great piece of work, but for tables 1 and 2, the authors should include cisplatin as a control, and discussing the findings as comparisons.
Author Response
Reviewer 2
A great piece of work, but for tables 1 and 2, the authors should include cisplatin as a control, and discussing the findings as comparisons.
Response:
Thank you for your constructive comment.
We have previously reported the tumor-specificity [TS=10.4(D/B)+4.6(C/A)=15; PSE=1551(100D/B2)+246 (100C/A2)=1797] [27] and neurotoxicity of cisplatin (B/E=0.73/4.4=0.17) [20]. Comparison with these values, the tumor-specificity of PB (TS=30.8+40.0=70.8; PSE=6722+13197=19917) (Table 1) was 4.7-fold (70.8/15) ~ 11.1-fold (19919/1797) that of cisplatin. On the other hand, the neurotoxicity of PB (B/E=2.0) was 11.8-fold that of cisplatin (B/E=4.4/0.73=0.17). This further support the notion that PB is the first class of both anticancer drug and neurotoxic agent, with these two activities tightly coupled.
This statement was added to the discussion (line 392-399).

Round 2
Reviewer 1 Report
The revised version is corrected, all comments approved. It looks much better, and now the manuscript is easy to read. I believe it will be interesting for the readers.